# Observation of non-reciprocal harmonic conversion in real sounds

Xinxin Guo [1], Hervé Lissek [1] & Romain Fleury [2✉]

Reciprocity guarantees that in most media, sound transmission is symmetric between two points of space when the location of the source and receiver are interchanged. This fundamental law can be broken in non-linear media, often at the cost of detrimental input power levels, large insertion losses, and ideally prepared single-frequency input signals. Thus, previous observations of non-reciprocal sound transmission have focused on pure tones, and cannot handle real sounds composed of various harmonics of a low-frequency fundamental note, as generated for example by musical instruments. Here, we extend the reach of non-reciprocal acoustics by achieving large, tunable, and timbre-preserved non-reciprocal transmission of sound notes composed of several harmonics, originating from musical instruments. This is achieved in a non-linear, actively reconfigurable, and non-Hermitian isolator that can handle arbitrarily low input power at any audible frequency, while providing isolation levels up to 30dB and a tunable level of non-reciprocal gain. Our findings may find applications in sound isolation, noise control, non-reciprocal and non-Hermitian metamaterials, and analog audio processing.

[1] Signal Processing Laboratory LTS2, Ecole Polytechnique Fédérale de Lausanne, 1015 Lausanne, Switzerland. [2] Laboratory of Wave Engineering, Ecole Polytechnique Fédérale de Lausanne, 1015 Lausanne, Switzerland. ✉email: romain.fleury@epfl.ch

Reciprocity is a fundamental principle of wave motion in linear, time-invariant, and time-reversible media, stating that the transmission coefficient between two points remains identical when the source and receiver positions are exchanged[1,2]. Such symmetry in the transmission is, however, not always desirable: for example, reciprocity breaking is a key ingredient in topological physics, allowing for the creation of chiral edge modes. More generally, large reciprocity breaking is required for unidirectional and perfectly controlled signal delivery when signal back-reflection is not desired. In acoustics, different strategies have led to large reciprocity breaking, realizing diverse non-reciprocal devices such as rectifiers[3–5], switches[6,7], diodes and isolators[7–12], circulators[13], topological insulators[14,15], and metamaterials with asymmetric functionalities[16–18]. Amongst all the developed approaches for eluding reciprocity, taking advantage of non-linear effects is arguably the most straightforward, requiring merely an appropriate integration of non-linearity with spatial asymmetry[1]. To date, many non-linear mechanisms have been used to achieve non-reciprocity, including higher harmonic generation[3,19], bifurcation[6,20,21] and chaos[22,23], solitons[24], acoustic radiation pressure[7], or other amplitude dependent responses[23,25–28], most of which rely on passive, intrinsically non-linear materials, resulting in no or very limited adjustment. Thus active control[29–37] appears to be an efficient means to provide more degrees of freedom.

Typically, non-linear non-reciprocal devices operate in ideal peculiar conditions that hinder their application to real sounds. First, the input power is often required to be large to efficiently trigger non-linearities[3,4,6,8,27]. Possible improvements proposed are either conceptual in strategy[25] or accessible at the cost of high design complexity[7,19,28]. Second, non-linear non-reciprocity is often limited to a narrow bandwidth, primarily explored with single-frequency tones[3,4,6,7,20,27]. However, real sounds, such as the notes produced by musical instruments, are typically composed of multiple harmonics spaced over a broad frequency range.

Here, we report the observation of giant non-reciprocal scattering of real multi-harmonic sounds. Notes originating from various musical instruments are up-converted by one octave when traveling in the forward direction without changing their timbre, whereas sound transmission in the opposite direction is prevented. This is achieved by active control of an electro-acoustic isolator that induces giant non-linear and non-Hermitian acoustic harmonic conversion, leading to large non-reciprocity even at low input intensities. Our experiment implements a fully adjustable and reconfigurable non-reciprocal harmonic converter of total length below $\lambda/4$ for audible frequencies spanning more than one octave. We demonstrate large non-reciprocal behavior on notes emitted by four different musical instruments. Our results open new possibilities for audio processing, sound isolation, non-Hermitian acoustics, and topological sonic materials with unusual acoustic behaviors.

## Results

**Design**. Consider the non-linear non-Hermitian electro-acoustic system depicted in Fig. 1. It combines a non-linear part that is capable of generating higher harmonics together with a linear part whose goal is to filter out a fundamental frequency $f$. The non-linear part is composed of a single actively controlled electro-acoustic resonator (named NL) involving a feedback loop on the front acoustic pressure $P_{NL}$, returning a drive current with a quadratic non-linearity adjustable by the constant $G_{NL}$. By increasing $G_{NL}$, one can create significant non-linearities while providing extra gain to the system, regardless of the input amplitude. The harmonics stimulated by such non-linear non-Hermitian control are considerably amplified when the resonance

also comes into play (see instances in Supplementary Fig. 2 in Supplementary Note 2). While the other resonators (named L1 and L2) are subject to linear controls, the feedback loops involve standard linear transfer functions $\Phi_{1,2}$, following the impedance control approach reported by Rivet et al.[38] (see the Method section and Supplementary Note 2 for more details). In the case of a wave incident at a 90° angle, the strategy required for frequency filtering is to minimize the resistance of the side resonator, resulting in the incoming energy being entirely reflected. In practice, however, this requires a pure reactive impedance, which is impossible to achieve by controlling a single resonator for stability reasons. That is why two active linear units, L1 and L2, are needed. The full system is a sub-wavelength acoustic liner with a total length of 20 cm, in which each of the controlled electrodynamic loudspeakers is enclosed in a 3D-printed cavity. Figure 1a, b corresponds to two transmission configurations, i.e., the forward configuration (Fig. 1a), where the incident wave comes and impinges first the unit NL and the backward configuration (Fig. 1b) is defined as the opposite. The principle of non-reciprocal conversion is to enable, in the forward direction, the transmission of higher harmonics mediated by the non-linear non-Hermitian resonator but disable it in the backward direction by linearly reflecting the fundamental tone.

**Non-reciprocal acoustic harmonic conversion**. We implemented the above-presented active controls in a compact prototype. Before considering multi-harmonic sounds, we start as an example with a pure tone excitation at 400 Hz. At this frequency, the total device length is $\approx 0.23\lambda$. Figure 2 shows the measured non-reciprocal transmission. The acoustic pressure input to the system is obtained from a calibration measurement made in the absence of the liner. It is measured to be about 1.3 Pa at 400 Hz when the source is fed by a 20 mVrms voltage source. Such a low input level is adopted to ensure that the non-linearity results only from the active control.

When the pure tone at 400 Hz is incoming along the forward direction, the first encountered unit cell NL, subject to quadratic non-linear control, generates higher harmonics, with the second harmonic taking precedence. The next two unit cells, L1 and L2, subject to linear impedance controls, allow the incident fundamental wave to be mostly reflected while higher harmonics pass through. Due to this reflection, the pressure in front of the unit NL ($P_{NL}$), located before L1 and L2, is enhanced, consistent with Fig. 2a, where the fundamental component of $P_{NL}$ goes up to 4.5 Pa. This, in turn, boosts the generation of higher harmonics. The final transmitted higher-frequency wave $P_{tr}$, after the fundamental wave is eliminated by L1 and L2, possesses an overall transmitted energy greater than that of the input and is mostly carried by the second harmonic. Conversely, in the backward configuration shown in Fig. 2b, the incident wave is directly reflected by the active linear units L2 and L1. The tiny residual amount of transmitted energy cannot trigger harmonic conversion at the last active unit cell NL, resulting in the wave transmission being prevented, with an amplitude as small as 0.04 Pa.

In order to characterize and quantify the energy transmitted in the two directions, we considered the Sound Intensity Level (SIL in decibels) defined as

$$\text{SIL} = 10\log_{10}\left(\sum_n |P_{tr}(n\omega)|^2\right), \quad (1)$$

where the summation goes over all generated harmonics of magnitude $|P_{tr}(n\omega)|$. A truncation to the first ten harmonics (until $n = 10$) is proved sufficient to calculate the SIL accurately. The difference in SILs between the two transmissions, a metric generalizing the notion of isolation ratio, is found to be as large as 30 dB, with a forward transmission gain of 1 dB.

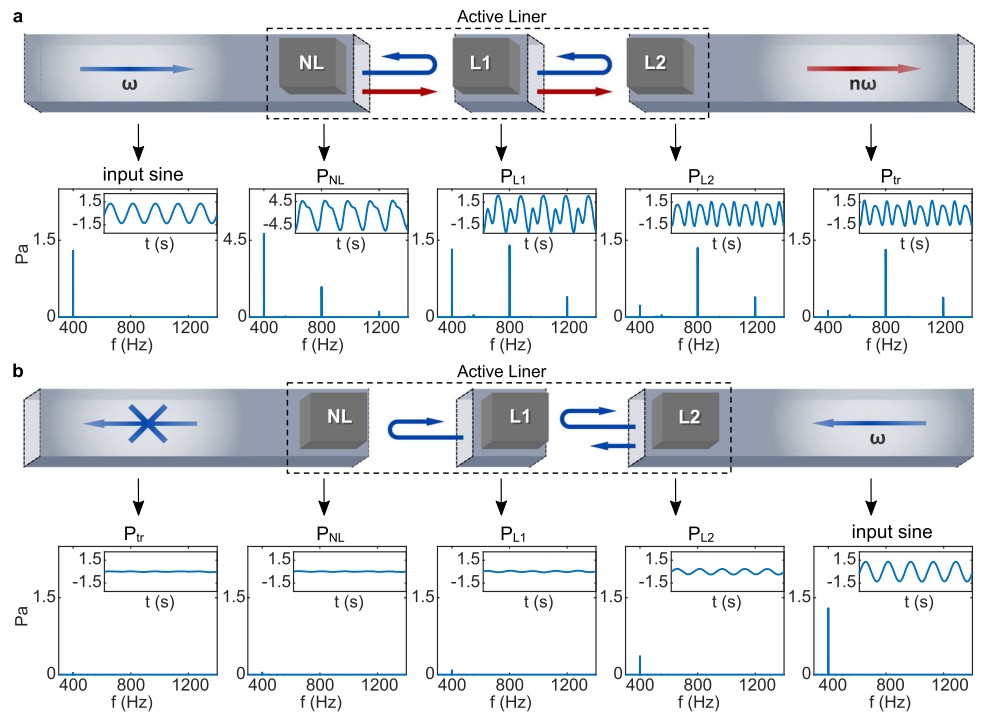

**Fig. 1 Principle of non-reciprocal acoustic harmonic conversion. a** Forward configuration, **b** Backward configuration. Active controls are defined by using the pressures ($P_{NL}$, $P_{L1}$, and $P_{L2}$) sensed in front of each cell (cell NL, L1, and L2) as the respective control inputs. Non-linear control is applied to the cell NL for higher harmonic generations, with $G_{NL}$ the constant gain to tune the level of non-linearity. Linear control is applied to the cells L1 and L2 for fundamental wave elimination, with $\Phi_1$ and $\Phi_2$ the transfer functions for the impedance adjustments (See the Method section and Supplementary Note 2 for details).

**Fig. 2 Non-reciprocal harmonic conversion process.** Wave propagation under the designed non-reciprocal isolator, with a pure tone excitation at 400 Hz. Evolution from the input pressure to that below each actively controlled unit cell ($P_{NL}$, $P_{L1}$, $P_{L2}$ below the cell NL, L1, L2, respectively) and then to the final transmitted wave ($P_{tr}$). Both the frequency spectrum and the time domain profile (insets) are displayed for all pressures in the **a** forward and the **b** backward configurations, respectively.

To explore the behavior of the system at other frequencies, we repeated the pure tone experiment in the range [180–475 Hz]. The control laws are reconfigured to enable harmonic conversion at different frequencies taken in this range with a step of 5 Hz. The measured SILs are summarized in Fig. 3a. The intensity of the input pressure is also shown as a reference.

For most frequencies, we observed a very large difference between the forward and backward SIL (similar results are found when considering wave-packet sources, as shown in Supplementary Note 5). We highlighted in orange the spectral bands for which this difference is larger than 20 dB. The frequency range between (370 and 440 Hz) shows very large non-reciprocal contrast, with an isolation ratio of up to 30 dB and with transmitted SIL even exceeding the input in 400–405 Hz. This enhancement is due to the strong reflection of the linear units, not to a resonant frequency conversion effect (see Supplementary

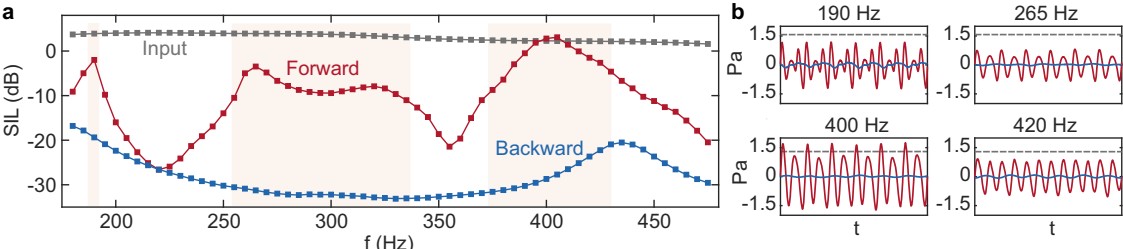

**Fig. 3 Experimental demonstration of non-reciprocal harmonic conversion with pure tones. a** Sound Intensity Level (SIL) of the transmitted wave ($P_{tr}$) in the forward and backward configurations, respectively, and compared to that of the input pressure. **b** Time domain waveforms of $P_{tr}$ in some representative cases, namely at 190, 265, 400, and 420 Hz, respectively, where the gray dashed lines indicate the input pressure level.

Note 3). Additionally, effective non-reciprocal transmission is also accessible in other frequency ranges, namely around 190 Hz and in the range of [255–330 Hz]. In these ranges, the transmitted wave is dominated by the third and the second harmonic, respectively, as exemplified by the few representative time signals plotted in Fig. 3b (corresponding spectra are in Supplementary Note 2). The prominence of these higher harmonics is attributed to a resonant harmonic generation process occurring when the resonant frequency of the NL resonator is a multiple of the excitation frequency (see proofs in Supplementary Notes 2 and 4). However, while the isolation ratios are high, the reflections from the active linear units (L1 and L2) lower the amplitude of the fundamental (Supplementary Note 3, Supplementary Fig. 3), making the transmitted wave in the forward direction lower than the input pressure, but with a moderate gap from 6 to 12 dB (corresponding to an intensity scale ratio from 0.5 to 0.25). Nevertheless, the observed non-reciprocal behavior is ideal for handling sounds composed of multiple harmonics of a fundamental tone, as we now demonstrate.

**Non-reciprocal harmonic conversions of real sounds**. Four instruments, namely the ocarina, clarinet, flute, and piano, are considered. The standard G4 notes from each instrument (MIDI note system), with fundamental frequencies around 392 Hz, are used as inputs. Figure 4 shows the measured transmitted sounds when using a linear control bandwidth of 40 Hz.

For all four instruments, the forward transmission through the active liner always converts the input G4 note into a G5 note, namely one octave up and with comparable or even higher volume. In contrast, the backward transmitted sound is significantly weaker. The G4 note of the ocarina exhibits the most homogeneous spectrum with mainly one frequency range of (370–410 Hz) being excited, which, as expected, can be almost completely blocked in the backward transmission. One of the clarinets presents a small third harmonic. Thus, the weak backward transmission corresponds to a D6 note (with a fundamental frequency three times that of a G4 note) but at a much lower volume. As for the other two instruments (the flute and the piano), their G4 notes excite simultaneously the frequency ranges around the second and the third harmonics, leading to conversion into G5 notes through backward transmission, similar to the forward configuration, albeit with a drastically smaller volume. It is also worth emphasizing that in transmissions along both directions, the distinctive timbre of each instrument is always preserved (refer to Supplementary Audios 1–4 for the recorded results of ocarina, clarinet, flute, and piano, respectively). This is in stark contrast with conventional non-linear acoustic isolators, that typically largely distort the incident signal.

## Discussion

We demonstrated giant non-reciprocal conversion of real multi-harmonic sounds in a non-linear, actively reconfigurable, and

non-Hermitian sonic isolator. Notes from four different musical instruments are up-converted by one octave in the forward direction and reflected in the other, while the unique timbre of each instrument is retained in both directions. Such large non-reciprocal harmonic conversion occurs over a distance smaller than a quarter wavelength. Isolation ratios up to 30 dB were attained in the frequency range [180–475 Hz], with the maximal contrast near 400 Hz, where the forward transmission rate even exceeds unity. This is attributed to non-linearity enhancement caused by the linear control-induced reflection or the frequency matching between a higher harmonic and the resonance.

Since we have broken reciprocity in a fully active manner through an assignment of feedback currents, our non-reciprocal device is mechanically robust yet without necessitating complex fabrication. Additionally, unlike the majority of previous studies that focused on normal incidence, our grazing incidence solution offers more flexibility for real-life applications, as the wave propagation is not physically obstructed, and the device cross-section is not constrained. While our study has already demonstrated a significant level of non-reciprocal harmonic conversion, it is possible to improve it further within a specific narrower frequency range, by superimposing the positive effects of resonance and linear wave scattering control. Moreover, since the non-linear converter is inherently broadband, combining more linear resonators (each with different but close resonances) can extend the effective bandwidth of the overall active devices, potentially making it possible to escape the trade-off between insertion loss, non-reciprocity, and bandwidth. Finally, our integration of non-linearity with active control may open new perspectives for experimental explorations at the intersection of non-linear physics and other fields, such as non-Hermitian[39,40] and topological acoustics[14,41,42].

## Methods

The three-unit cells used are all made of commercially available loudspeakers Visaton FRWS 5 in their individual enclosures (refer to Supplementary Note 1 for their photos and Thiele/Small parameters). When subjected to a sufficiently weak external acoustic pressure $p_f(t)$ at the front face, each closed-box loudspeaker behaves as a linear single-degree-of-freedom resonator. To better understand how the controls are implemented, we first describe the linear dynamics of the loudspeaker membrane in the time domain as follows:

$$M_{ms}\frac{dv(t)}{dt} = S_d p_f(t) - R_{ms}v(t) - \frac{1}{C_{mc}}\xi(t)dt - B\ell i(t), \quad (2)$$

where $v(t)$ and $\xi(t)$ denote the axial inward velocity and displacement of the diaphragm, respectively. The resonance properties exhibited are related to the mechanical and electromagnetic parameters of the resonator actuated, namely the effective surface area and the moving mass of the diaphragm $S_d$ and $M_{ms}$, the force factor of the moving coil $B\ell$, the resistance factored into the global losses $R_{ms}$, and the overall equivalent compliance $C_{mc}$ where both the elastic suspension of the loudspeaker and the compressibility of the fluid inside the enclosure are accounted for (details can be found in Rivet[38] or Guo et al.[43]).

The active controls developed here act on the electrical current $i(t)$ circulating in the loudspeaker moving coil, and are undertaken with an FPGA-based Speedgoat performance real-time target machine operated by the xPC target environment of MATLAB (SIMULINK). Taking the sensed front pressure $p_f(t)$ as the input signal, $i(t)$ is specified through a digital definition of control laws (see details in

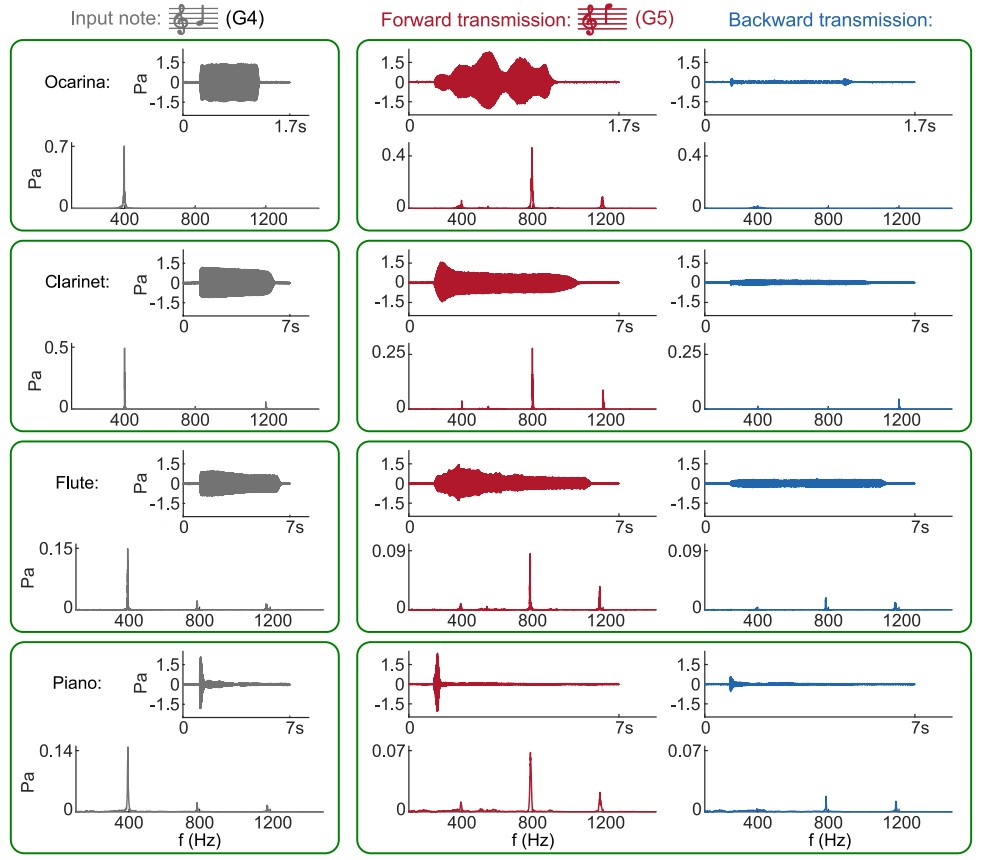

**Fig. 4 Experimental demonstration of non-reciprocal harmonic conversion with real sounds.** Achievement with standard G4 notes (MIDI note system) from four instruments, namely ocarina, clarinet, flute and piano. The respective time-domain results (input note, forward and backward transmitted notes) are recorded in Supplementary Audios 1 to 4, respectively.

Supplementary Note 2), the output (voltage) of which is converted into a current through a custom-made current-drive amplifier (op-amp based improved Howland current pump circuit[32]). The acoustic properties of the controlled resonator are customized by feeding back such a defined current in real time.

Regarding the linear impedance control approach applied to the units L1 and L2, it uses a transfer function $\Phi$ to tailor the impedance of the resonator, more specifically, to prescribe its mechanical resistance $R_{ms}$, overall compliance $C_{mc}$, and moving mass $M_{ms}$. For L1 and L2, these parameters are altered toward the same target acoustic impedance (same resonance frequency and magnitude). The frequency selectivity of the linearly controlled liner is assessed through the transmission coefficient measured with the standard transfer matrix method, using the 4 microphones mic 1 to mic 4 drawn in Fig. 1. The resistance elimination controls required on L1 and L2 are determined such that $|T| \leq 0.03$ is reached at each target frequency interval. The compliance adjustments are performed for shifting the resonance, the moving mass is tuned depending on the effective bandwidth in need (see details of the linear control approaches in Supplementary Note 2, with examples of pure linear control results in Supplementary Fig. 2).

All the pressures are sensed with PCB Piezotronics Type 130D20 ICP microphones. They present a rather flat response in the whole working frequency range. The gains (sensibilities) they added to the pressures are compensated for in the definition of the control laws. The PVC duct used has a square cross-section of 5 cm × 5 cm to ensure plane wave propagation until 3400 Hz. An anechoic termination (with melamine foam of appropriate shape filled inside) is put into place at the end of the duct to provide a reflection-free boundary condition (absorption coefficient greater than 0.998) from 180 Hz (see Supplementary Fig. 1 in Supplementary Note 1 for setup picture). As such, the actual transmitted wave is directly available from the total pressure sensed by the microphone close to the end (we choose the closest one, i.e., mic 4 in the forward configuration, and mic 1 in the backward configuration). In presence of non-linearity, all the measurements are performed in the time domain, from which the desired harmonic distributions are extracted through a Fourier transform.

## Data availability

The experimental data are available from the corresponding author upon reasonable request.

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

## Acknowledgements

This research is supported by the Swiss National Science Foundation (SNSF) under grant No. 200020_200498.

## Author contributions

X.G. conceived, designed, and performed the experiments. R.F. and H.L. supervised this work. All authors contributed to the interpretation of the results and the writing of this paper.

## Competing interests

The authors declare no competing interests.
