## [Peer review file · Communications Physics]

Observation of non-reciprocal harmonic conversion in real soundsReviewer #1 (Remarks to the Author):

see attachment

Reviewer #2 (Remarks to the Author):

The authors have proposed and realized an nonlinear and non-Hermitian system for unidirectional frequency harmonic conversion. The system is constructed by cascading a nonlinear element in doing frequency conversion with a linear element in blocking fundamental frequency. Then high harmonics can be generated when the system is excited from one direction with elimination of fundamental frequency while just a blocking effect from another direction.

The work is interesting as well as useful for future developments. Specifically, it shows that artificial non-linearity can be made extremely large in such a setting. As a result, more than 20dB difference between the forward and backward direction is found as non-reciprocal control. The manuscript is very well written as well. I proposed it can be accepted as is. I have some suggestions that can be considered.

Please comment on whether there can be some control on the proportions between the different harmonics transmitted from one side.

Some recent developments on artificial nonlinearity *New J. Phys.* 21 (2019) 093012 and active non-Hermitian acoustic metamaterials for frequency conversion *Commun. Physics* 5,18 (2022) can be cited.

The manuscript discussed a design of non-reciprocal acoustic harmonic convertor for airborne sound consisting of three active loudspeakers with one connected to a nonlinear circuit and the other two connected with linear controllers. The working principle is similar to the passive design combining nonlinear medium and bandgap crystal reported in *Nature Mater.* (2010) and the partially active design containing nonlinear circuit and asymmetric cavities reported in *Nature Comm.* (2014). Because of the purely active design of this work, the convertor may have the potential to achieve nonreciprocity over a broad frequency band in the audible range. The manuscript is well-written, and the results are interesting. I have three major concerns that need to be addressed before the manuscript can be considered for publication.

1. In Fig. 3 of the manuscript, it showed that the convertor has bandgaps between 190 Hz to 255Hz, 340 Hz to 370 Hz, and above 430 Hz. While this convertor is a purely active design, what is the cause of these bandgaps? Are these bandgaps caused by the resonances of the loudspeakers or control circuits? If so, can the loudspeakers or circuits be replaced to achieve broadband response? If not, the narrow-band feature of this design will make it similar to the previous passive and partially active designs in terms of functions that will significantly diminish the novelty and limit the applications of this work.
2. Even though that the authors claimed the modulation of real sound and used musical instruments for their experiments, pure tone with narrow bandwidth sound was used. Will broadband sound or a complete song be distorted by the nonreciprocal convertor? This concern is also related to the frequency band limitation of the design that will diminish the novelty and limit the applications of the design.
3. In the Supplementary Fig. 2a, the sketch of the loudspeaker and control circuits includes a microphone to measure the pressure wave and form a complete feedback loop. However, the microphone was not mentioned anywhere else in the manuscript or considered in their modeling. There may be a scientific flaw in their theoretical modeling without considering the response of the microphone and the feedback control.

Comments of Reviewer 1:

The manuscript discussed a design of non-reciprocal acoustic harmonic convertor for airborne sound consisting of three active loudspeakers with one connected to a nonlinear circuit and the other two connected with linear controllers. The working principle is similar to the passive design combining nonlinear medium and bandgap crystal reported in *Nature Mater.* (2010) and the partially active design containing nonlinear circuit and asymmetric cavities reported in *Nature Comm.* (2014). Because of the purely active design of this work, the convertor may have the potential to achieve nonreciprocity over a broad frequency band in the audible range. The manuscript is well-written, and the results are interesting. I have three major concerns that need to be addressed before the manuscript can be considered for publication.

1. In Fig. 3 of the manuscript, it showed that the convertor has bandgaps between 190 Hz to 255Hz, 340 Hz to 370 Hz, and above 430 Hz. While this convertor is a purely active design, what is the cause of these bandgaps? Are these bandgaps caused by the resonances of the loudspeakers or control circuits? If so, can the loudspeakers or circuits be replaced to achieve broadband response? If not, the narrow-band feature of this design will make it similar to the previous passive and partially active designs in terms of functions that will significantly diminish the novelty and limit the applications of this work.

We thank the reviewer for the relevant question. Indeed, in order to enhance the generation of higher harmonics, we made use of either (i) the linear control-induced reflection to enhance the fundamental wave (as happened around 400 Hz), or (ii) the matching between one higher harmonic and the resonance (achieved in the two other effective ranges). The band gaps correspond to the frequency ranges where none of these two effects can come into play. As the reviewer noticed, these band gaps are closely related to the resonance of the loudspeakers. In this view, it would be quite straightforward to remove these bandgaps by adding another loudspeaker with a different resonance frequency, since the non-reciprocal harmonics are anyways always transmitted through the initial system. Thus, our active control approach is not fundamentally limited to a non-continuous broad range of frequencies, the price to pay for continuous bands is extra complexity. We included a

remark about this in the main text.

2. Even though that the authors claimed the modulation of real sound and used musical instruments for their experiments, pure tone with narrow bandwidth sound was used. Will broadband sound or a complete song be distorted by the nonreciprocal converter? This concern is also related to the frequency band limitation of the design that will diminish the novelty and limit the applications of the design.

The limitation of the frequency band results from the linear control on the loudspeakers, since instability occurs when a larger bandgap is required. That is why with only two linearly controlled loudspeakers, we cannot achieve a continuously broadband device. However, such limitation can be improved simply by implementing more active linear resonators. Since each linearly controlled loudspeaker can create a bandgap of around 40Hz (as demonstrated in the manuscript), combining several of them (each targeting a different but close resonance) allows for broadband isolation. The proposed pure active device is undoubtedly more efficient than any passive structure, since achieving the same effect with fewer units.

With a view to showing the potential of our active control approach, in the last paragraph of the discussion part, we have modified the following sentence:

'Moreover, since the non-linear converter is inherently broadband, combining more linear resonators (each with different but close resonances) can extend the effective bandwidth of the overall active devices, potentially making it possible to escape the trade-off between insertion loss, non-reciprocity and bandwidth.'

3. In the Supplementary Fig. 2a, the sketch of the loudspeaker and control circuits includes a microphone to measure the pressure wave and form a complete feedback loop. However, the microphone was not mentioned anywhere else in the manuscript or considered in their modeling. There may be a scientific flaw in their theoretical modeling without considering the response of the microphone and the feedback control.

We apologize if the use of microphones was not evident in the initial draft.

They were already illustrated in the sketch of Fig.1 of the main manuscript (white circle blocks in front of the loudspeakers), albeit not discussed very explicitly in the main text. We now make sure that the description of the feedback control schemes is given explicitly in the first paragraph of section 'Results'. We have also corrected Fig.1 by better highlighting these microphones (with brown blocks and a legend) and also adding an explicit description in the caption.

Regarding the technical part of your comment, the quarter-inch microphones we used are first-class PCB Piezotronics Type 130D20 ICP microphones. They present a very flat frequency response up to a few kHz, and can thus safely be modeled as dispersionless gains that convert pressure to voltage. To avoid any misunderstanding, we have added the description in the section 'Method'.

Comments of Reviewer 2:

The authors have proposed and realized an nonlinear and non-Hermitian system for unidirectional frequency harmonic conversion. The system is constructed by cascading a nonlinear element in doing frequency conversion with a linear element in blocking fundamental frequency. Then high harmonics can be generated when the system is excited from one direction with elimination of fundamental frequency while just a blocking effect from another direction.

The work is interesting as well as useful for future developments. Specifically, it shows that artificial non-linearity can be made extremely large in such a setting. As a result, more than 20dB difference between the forward and backward direction is found as non-reciprocal control. The manuscript is very well written as well. I proposed it can be accepted as is. I have some suggestions that can be considered.

1. Please comment on whether there can be some control on the proportions between the different harmonics transmitted from one side.

We think it is possible to control the proportions between the different harmonics transmitted. In our work, it is quite straightforward to tailor the transmitted higher harmonics. There is a direct link between (i) the control parameter G_{NL} that we used for tuning the level of nonlinearity and

(ii) the magnitude of the transmitted higher harmonics. Since we consider a quadratic nonlinearity, as G_{NL} increases, the second harmonic is the first being excited and then the third one. Therefore, we can modify the ratio between higher harmonics by tuning G_{NL} . We can also take advantage of resonance to increase significantly one of the harmonics excited, as we did in our work. Another possible solution is to change the nonlinear control law as needed. For instance, we can combine the quadratic nonlinearity with a cubic one, and assign each of them a constant control gain to tune the second and the third harmonics independently.

2. Some recent developments on artificial nonlinearity *New J. Phys.* 21 (2019) 093012 and active non-Hermitian acoustic metamaterials for frequency conversion *Commun. Physics* 5,18 (2022) can be cited.

We thank the reviewer for the updates, we have added these two papers in the reference and cited them in the manuscript.

REVIEWERS' COMMENTS:

Reviewer #1 (Remarks to the Author):

The authors have addressed my concerns and the manuscript is good for publication.